# Enhanced diffusion by binding to the crosslinks of a polymer gel

Carl P. Goodrich [1,2], Michael P. Brenner[1,2] & Katharina Ribbeck[3]

Creating a selective gel that filters particles based on their interactions is a major goal of nanotechnology, with far-reaching implications from drug delivery to controlling assembly pathways. However, this is particularly difficult when the particles are larger than the gel's characteristic mesh size because such particles cannot passively pass through the gel. Thus, filtering requires the interacting particles to transiently reorganize the gel's internal structure. While significant advances, e.g., in DNA engineering, have enabled the design of nano-materials with programmable interactions, it is not clear what physical principles such a designer gel could exploit to achieve selective permeability. We present an equilibrium mechanism where crosslink binding dynamics are affected by interacting particles such that particle diffusion is enhanced. In addition to revealing specific design rules for manufacturing selective gels, our results have the potential to explain the origin of selective permeability in certain biological materials, including the nuclear pore complex.

[1] School of Engineering and Applied Sciences, Harvard University, Cambridge, MA 02138, USA. [2] Kavli Institute of Bionano Sciences and Technology, Harvard University, Cambridge, MA 02138, USA. [3] Department of Biological Engineering, Massachusetts Institute of Technology (MIT), Cambridge, MA 02139, USA. Correspondence and requests for materials should be addressed to C.P.G. (email: goodrich@g.harvard.edu)

Hydrogels composed of crosslinked hydrophilic polymers have an important role as selective permeability barriers in regulating the diffusive transport of molecules in a wide variety of biological systems, with examples ranging from cartilage to mucus to the extracellular matrix[1–3]. These represent a class of materials that allow the passage of certain molecules while rejecting other. However, despite the obvious technological importance of designer microscopic filtering devices, our understanding of the physical mechanisms that could enable such technology is surprisingly underdeveloped aside from a few simple and limited cases.

There are two well-understood ways that certain particles can be retained by a polymer-based network while other particles are able to pass through. First, particles are retained when they are larger than the characteristic mesh size of the gel (Fig. 1a)[1,4–9], which is the typical spacing between the polymers within the gel; particles smaller than the mesh size are able to pass through the gel, diffusing with a diffusion constant limited by the local viscosity. On the time scale of structural reorganization within the gel, even large particles can diffuse, but this is typically much slower than the diffusion of small particles, making this an effective mechanism for differentiation between large and small particles. Second, many biological systems (e.g., mucus, the extracellular matrix) use non-steric interactions to distinguish between particles of similar size[1,10–13]. For example, particles that are smaller than the mesh size can still be retained by binding directly to the polymers of the gel (Fig. 1b).

In this binding-mediated mechanism, particles facilitate their own retention: generic particles diffuse through the gel while specific binding particles are trapped. Can a particle use binding interactions to instead facilitate its own diffusion? Such a mechanism is seemingly paradoxical because binding typically suppresses motion rather than enhancing it. However, in the nuclear pore complex, which is a large protein pore with a gel-like plug that regulates the transfer of macromolecules between the nucleus and the cytoplasm, certain transporter molecules that are larger than the mesh size are observed to facilitate their own diffusion through the plug without spending energy[1,14–19]. It is believed that this facilitation is mediated through specific hydrophobic binding interactions, but a microscopic mechanism for this class of facilitated diffusion is not understood[1,19–35].

In this paper, we present a physical mechanism where particles facilitate their own diffusion by binding to a gel, leading to enhanced diffusion compared to otherwise equivalent non-binding particles. This mechanism applies to particles that are larger than the characteristic mesh size of the gel, so non-binding particles are caged by the surrounding polymers. Binding particles, however, are able to alter the local structure of the gel, which enables diffusion.

## Results

**Crosslinking and crosslink dynamics.** Since large particles are caged by the surrounding polymer gel, we seek a mechanism where binding allows large particles to efficiently escape their cage. One possible approach is for binding to facilitate polymer degradation (Fig. 2a). However, permanently destroying the nearby polymers affects the structural integrity of the gel and reduces the capacity for future selectivity. Instead, we will achieve specificity by allowing binding particles to escape their cage by breaking crosslinks, resulting in a fully healable and reversible system. For concreteness going forward, we consider gels with two types of complementary binding sites, denoted $G$ and $G'$, which can come together to form $GG'$ crosslinks.

How can a particle cause a crosslink to break? Our mechanism relies on two key features. The first is competitive binding at the crosslink binding sites, where the particle binds directly to the $G$ crosslink binding sites (as opposed to a random location on the polymer) so that each $G$ binding site cannot be both crosslinked and bound to a particle simultaneously. Thus, each $G$ site can be in one of three states (Fig. 2b): (1) free, (2) crosslinked, i.e., attached to a $G'$ binding site to form a $GG'$ crosslink, or (3) bound to a particle to form a $GP$ bond.

Competitive binding allows binding particles to reduce the local concentration of crosslinks. Assuming arbitrary transition rates (see below) the equilibrium probability of an individual $G$ binding site being in the crosslinked state is

$$P_{\text{crosslinked}} = \frac{c_{\text{eff},G'}k_{\text{fc}}}{k_{\text{cf}} + c_{\text{eff},G'}k_{\text{fc}} + c_{\text{eff},P}k_{\text{cf}}k_{\text{fb}}/k_{\text{bf}}}, \quad (1)$$

where $c_{\text{eff},G'}$ and $c_{\text{eff},P}$ are the local effective concentrations of the $G'$ binding sites and binding particles, respectively, in the near vicinity of the $G$ binding site in question. Since the last term in the denominator vanishes when $c_{\text{eff},P} = 0$ (i.e., when there is no nearby diffusing particle or when the particle is non-binding) and all rates are non-negative, binding reduces $P_{\text{crosslinked}}$. However, binding also reduces the probability of being in the free state,

$$P_{\text{free}} = \frac{k_{\text{cf}}}{k_{\text{cf}} + c_{\text{eff},G'}k_{\text{fc}} + c_{\text{eff},P}k_{\text{cf}}k_{\text{fb}}/k_{\text{bf}}}, \quad (2)$$

for exactly the same reason.

The key idea is to note that enhanced diffusion requires the binding particle to decrease the lifetime of nearby crosslinks, not the total number of nearby crosslinks. This requires that binding sites transition directly from the crosslinked $GG'$ state to the bound $GP$ state without passing through the free $G$ state (see the arrows in Fig. 2b). This direct bond exchange makes the lifetime of the crosslinked state

$$\tau_{\text{crosslinked}} = \frac{1}{k_{\text{cf}} + c_{\text{eff},P}k_{\text{cb}}}, \quad (3)$$

and is the second key feature of our model. This lifetime clearly decreases when $c_{\text{eff},P} > 0$, thus, providing a pathway for crosslinks to break that is only available in the presence of a binding particle.

Assuming equilibrium dynamics, the transitions between the free, crosslinked and bound states are governed by the three state energies, $E_{\text{f}}$, $E_{\text{c}}$, and $E_{\text{b}}$, and three transition state energies, $T_{\text{fc}}$, $T_{\text{cb}}$, and $T_{\text{bf}}$ (Fig. 2c). Since only energy differences affect transition dynamics, we can set $E_{\text{f}} = 0$, so the system is controlled

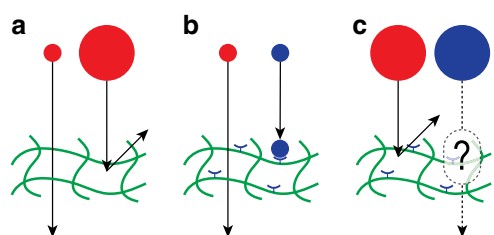

**Fig. 1** Three strategies for filtering particles through a polymer hydrogel. **a** From simple steric interactions, particles that are smaller than the mesh size of the gel are able to diffuse through the gel, while larger particles are retained. **b** While generic small particles diffuse through the gel, specific binding sites on the gel can slow down or trap non-generic particles that they bind to. This provides a simple mechanism to filter between equivalently sized small particles. **c** Since generic large particles do not diffuse through a gel, in order for a binding-based mechanism to successfully filter between equivalently sized large particles, binding must lead to enhanced mobility. How is this possible?

by only five variables. The only other a priori constraint on the parameters is that the transition state energies are greater than or equal to the energies of the two neighboring states (*e.g.* $T_{cb} \geq E_c$ and $T_{cb} \geq E_b$). These energies are sufficient to calculate the transition rates, so the rate of transitioning from state $\mu$ to state $\nu$ is[36]

$$k_{\mu\nu} \sim e^{-\left(T_{\mu\nu} - E_\mu\right)/k_B T},\qquad(4)$$

where $k_B T$ is the temperature times the Boltzmann constant. Note that the model satisfies detailed balance, so that $k_{fc}k_{cb}k_{bf} = k_{fb}k_{bc}k_{cf}$; the dynamics can be described either by the six transition rates combined with the detailed balance constraint, or through the five non-zero energies $E_c$, $E_b$, $T_{fc}$, $T_{cb}$, and $T_{bf}$. Note that the local effective concentrations $c_{eff,G'}$ and $c_{eff,P}$ depend on the dynamics of the particle and the gel. With the model so posed, the question is whether there exists a combination of state and transition state energies so that a binding particle will experience enhanced diffusion compared to a non-binding particle, whose diffusion is strongly suppressed due to its size.

**Perfect filtering with high binding affinity.** To see whether this model leads to enhanced diffusion of the binding particles, we carry out Brownian Dynamics simulations of a particle diffusing through a polymer gel (Fig. 3a, see Methods). We first consider the case of high binding affinity, where the energy barrier for spontaneously breaking a crosslink is large (i.e., $T_{fc} - E_c \gg k_B T$). In this limit, crosslinks do not break on their own so non-binding particles cannot diffuse. Thus, any mechanism that allows a binding particle to diffuse at all would result in a perfect filter. We find that such a mechanism can indeed be realized if the binding affinity of a *GP* bond is similarly high ($E_c \approx E_b$) with a small barrier between the bound and crosslinked states ($T_{cb} - E_c \approx k_B T$). This allows easy transitions back and forth between the

crosslinked and bound states, even though neither can transition to the free state (see the inset in Fig. 3b).

To show that this does indeed lead to a perfect filter, we run Brownian Dynamics simulations of a single binding or non-binding particle in a highly crosslinked gel, as discussed above and in the Methods section. Figure 3b shows the mean squared displacement of the particles as a function of time for both particle types (see Supplementary Movie 1). For non-binding particles (red data), the mean squared displacement quickly plateaus, indicating permanent caging. In contrast, the mean square displacement for binding particles (blue data) is linear at long times, indicating that it has reached the diffusive regime.

The mechanism for diffusion is depicted in Pathway 1, Fig. 3c. Initially, a binding particle starts off in a cage. As the particle diffuses in the cage, it eventually collides with one of the bounding crosslinks and the *G* binding site transitions, with some probability, from the crosslinked to the bound state. This breaks the crosslink, allowing the particle to diffuse outside of its original cage while remaining bound to the *G* binding site. Eventually, the *G'* binding site diffuses back, causing the transition back to the *GG'* crosslinked state and releasing the particle. At this point, there is a 50% chance that the particle is outside the original cage and thus has moved a short distance. As this process repeats itself, the particle hops from cage to cage, resulting in long-time diffusion. The mechanism thereby allows particles that bind to the gel to effectively slip through the crosslinks, whereas non-binding particles are caged indefinitely.

In this limit of high binding affinity, the only relevant energies are $E_c$, $T_{cb}$, and $E_b$ because they are much lower than the other three. Therefore, there are two relevant parameters: the difference in binding energies between the crosslinked and bound states, and the height of the barrier between the two states. Since the barrier height must be greater than or equal to both the neighboring states, we can parameterize these as $E_b - E_c$ and $T_{cb} - \max(E_c, E_b)$, respectively. Figure 4a shows the diffusion

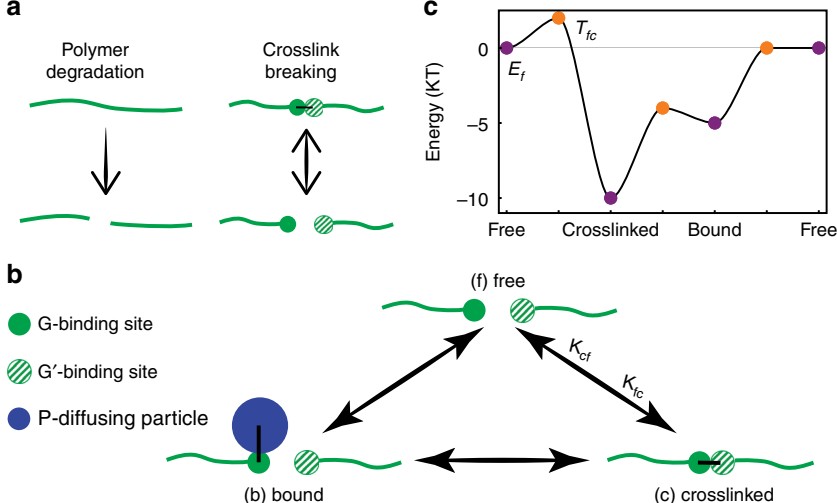

**Fig. 2** Model for crosslink dynamics. **a** Polymer degradation and crosslink breaking are two potential ways for a large particle to escape its cage in a polymer gel. However, crosslink breaking is reversible while polymer degradation is not. Therefore, our mechanism focuses on controlling the crosslink dynamics. **b** The *G* crosslink binding sites (solid green circles) can be in one of three states: free, where it is not bound to anything; crosslinked, where it is bound to a complementary *G'* crosslinking site (striped green circles); and bound, where it is bound to a binding particle (blue circle). Arrows show the transitions between the states, the dynamics of which are controlled by the three state energies ($E_f$, $E_c$, and $E_b$, respectively), and the three transition state energies ($T_{fc}$, $T_{cb}$, and $T_{bf}$). Without loss of generality, we set $E_f = 0$, but the remaining five energies are free parameters in our model. The transition rates (e.g., $k_{fc}$, the rate of transitioning from the free to crosslinked states) are given by the energies according to (4). Note that transition state energies must be greater than or equal to both neighboring state energies. **c** Plot of arbitrarily chosen state energies (purple) and transition state energies (orange). Since the reactions shown in **b** form a single closed loop, the reaction coordinate in **c** is periodic, as indicated by the repetition of the free state

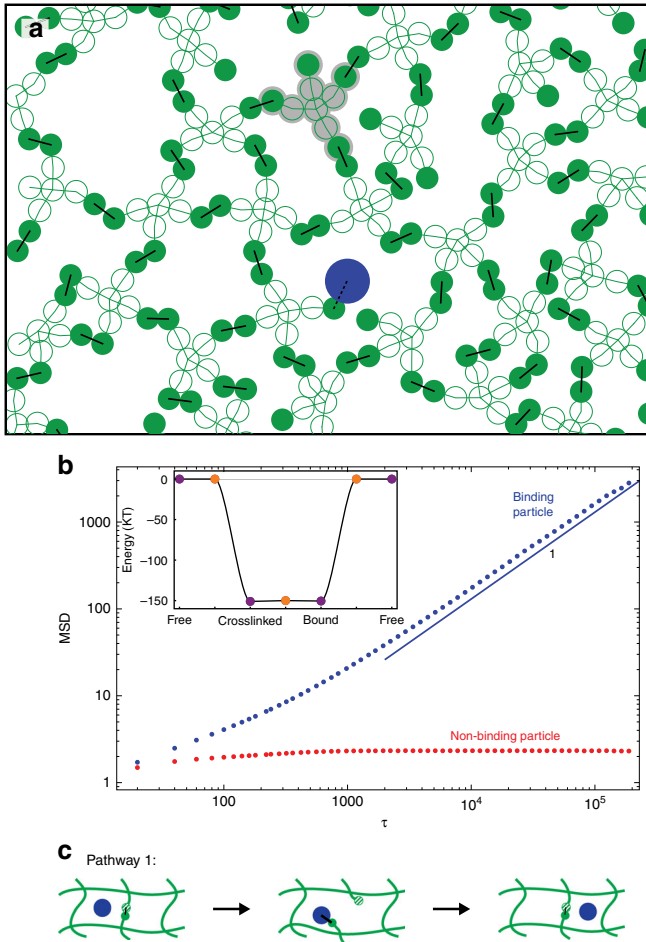

**Fig. 3** Perfect filtering in a gel with high binding affinities. **a** A diffusing particle (blue) and the surrounding gel (green). The gel is composed of star-shaped polymers that have a central particle connecting 4 polymer strands each of length 2 (the light green lines show permanent bonds within the polymers). One star-shaped polymer is shaded gray for clarity. The strand ends (solid circles) can be in one of three states: (1) free, (2) crosslinked with another strand end (solid black lines), or (3) bound to a particle (dashed black line). **b** Mean squared displacement of a binding particle (blue) and a non-binding particle (red) obtained from Brownian Dynamics simulations. Since the crosslinks are permanently bound, the non-binding particle is caged indefinitely and does not diffuse. However, the binding particle is able to escape its cage, and as a result has a non-zero diffusion constant. See Supplementary Movie 1. Inset: state and transition state energies used in these simulations. For these simulations, $E_f = T_{fc} = T_{bf} = 0$, while $E_c = -150.5 k_B T$ and $T_{cb} = E_b = -150 k_B T$. **c** A sketch of how a binding particle can diffuse through a gel by breaking crosslinks. Note that the binding site on the gel (solid green disk) is never in the free state, but instead transitions directly between the crosslinked and bound states. In this way, the particle can escape the cage while the total energy remains (roughly) constant since the total number of bonds in the system is conserved

constant, $D$, of the binding particle as a function of these two parameters, divided by the diffusion constant, $D_0$, of a free particle in the absence of any gel. As anticipated, diffusion is maximized when the three energies, $E_c$, $T_{cb}$, and $E_b$ are roughly equal to maximize diffusion.

Finally, Fig. 4b shows this mechanism being used as an actual filter. Instead of tracking the diffusion of a single particle, the gel is set up in a shell-like configuration with a distinct interior and exterior. Initially, 10 binding particles (blue) and 10 non-binding particles (red) are placed in the interior region (left panel). As time progresses (right panel), the binding particles are able to enter the gel, diffuse through it, and exit into the exterior region—effectively moving down the initial concentration gradient. However, all 10 non-binding particles remain in the interior (see Supplementary Movie 2).

Note that this mechanism works because the total number of bonds is exactly conserved (Fig. 3c). If a binding particle were to encounter a G binding site in the free state, the two could bind with no direct path for them to unbind, causing the particle to become trapped. Therefore, this mechanism only works when the gel is saturated, so that there are no free G binding sites in the gel. Another difficulty when considering this high affinity limit is that $E_c$ and $E_b$ must be matched to within a few $k_B T$ despite the magnitude of the energies being considerably larger. Whether in natural or artificial gels, it is not always clear how such precise control of the binding energies can be obtained.

**Enhanced diffusion with moderate binding affinity**. The qualitative mechanism outlined above can also work when all energy barriers are on the order of a few $k_B T$ so that all three states in Fig. 2b are accessible on reasonable time scales. Unlike the previous case, this means that crosslinks can spontaneously break and reform, resulting in a reversible gel. The gel structure evolves over a self-healing time scale, $\tau_h$. Therefore, even a non-binding particle will not be permanently caged but will diffuse with a diffusion constant $D_{non-binding} \sim \ell^2/\tau_h$, where $\ell$ is the mesh size of the gel. This process is shown in Pathway 2 in Fig. 5a. The particle starts in the same cage as before, but eventually one of the crosslinks will spontaneously break. This allows the particle to diffuse into the neighboring space, so that when the crosslink reforms (or new crosslinks are formed with other nearby binding sites), there is some probability that the particle will be in a new cage. As this process repeats itself, the particle will experience long-time diffusion, as confirmed numerically by the red data in Fig. 5c.

Can particle binding to the gel increase the particle diffusivity? Pathway 2 is clearly also available to a binding particle, as is Pathway 1 (Fig. 3c). However, there is also a third pathway available, Pathway 3, depicted in Fig. 5b. Here, the crosslink uses the binding particle as a catalyst to help it transition to the free state. Thus, the crosslink binding site passes through an intermediate bound state. Once the crosslink is in the free state, the particle is free to diffuse and has a chance to be in a new cage when the crosslink reforms. Having extra pathways for diffusion does not in itself imply that the binding particle diffuses faster than the non-binding particle. Nevertheless, we can indeed tune the binding and transition state energies so that $D_{binding} > D_{non-binding}$. A specific example is shown in Fig. 5c (see also Supplementary Movie 3), where we set $E_c = 2E_b = -10 k_B T$ and all transition state energies equal to the larger of the two adjoining states (see inset). The resulting mean squared displacement corresponds to a fivefold increase in the particle diffusivity. Figure 5d shows how the relative diffusivity $D_{binding}/D_{non-binding}$ varies when the different energy scales are varied from the simulation of Fig. 5c. There is a wide parameter range where particle binding facilitates diffusion. The most sensitive energy scale for increasing the effectiveness of the filter is $T_{fc}$; this suppresses the diffusion of the non-binding particle and extrapolates to the case of high binding affinity discussed above.

Note that all simulations described thus far focus on two-dimensional gels. As discussed in the Methods section, this provides a cleaner means to test the mechanism because in two

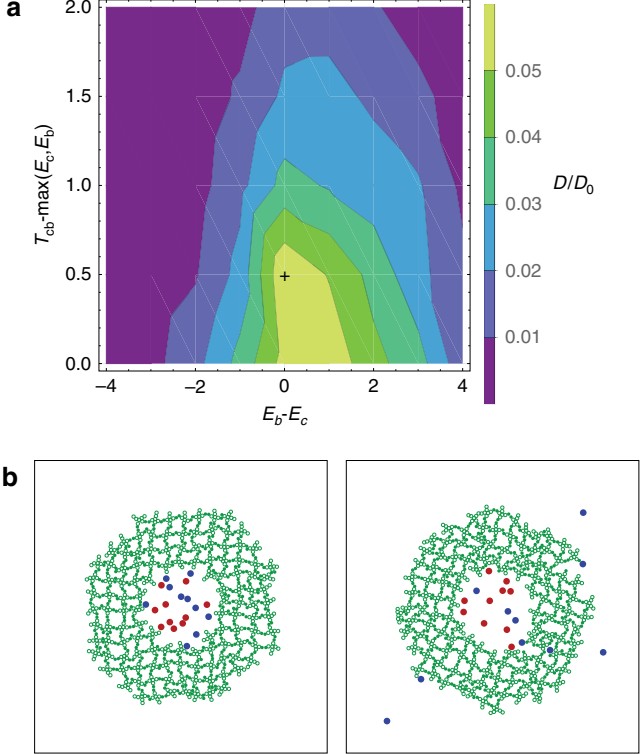

**Fig. 4** Analysis of filtering mechanism in high affinity limit. **a** Diffusion constant as a function of $E_b - E_c$ and $T_{cb} - \max(E_c, E_b)$, divided by the diffusion constant of a completely free particle with no gel. Since the energy barriers to spontaneous bond breaking are effectively infinite, this represents a complete search of the parameter space. The largest diffusion constant obtained is $D/D_0 = 0.059$ and is marked with a plus symbol. Data are averaged over 20 simulations for each point. **b** Demonstration of perfect filtering. Initially (left panel), a gel is trimmed to form a circular shell. 10 binding (blue) and 10 non-binding (red) particles are all placed within this shell. Over time (right panel), the binding particles are able to enter the gel and diffuse through it, while the non-binding particles remain in the interior. See Supplementary Movie 2

dimensions particles cannot passively diffuse simply because they are not sufficiently large. Nevertheless, we have also simulated the model in three dimensions to verify that our results are not dependent on dimensionality. Figure 6, which shows the mean squared displacement for binding and non-binding particles with the same parameters as in Fig. 5c, confirms that the large binding particle experiences enhanced diffusion, and therefore the mechanism proposed here is not restricted to two dimensions. The quantitative effect is less than in two dimensions (in this case $D_{\text{binding}}/D_{\text{non-binding}} \approx 1.5$), but this is expected because the mesh size is smaller compared to the particle size, so more crosslinks need to break to facilitate diffusion. An in-depth exploration of parameter space is beyond the scope of the current work, but we anticipate that $D_{\text{binding}}/D_{\text{non-binding}}$ can be highly optimized relative to the data shown in Fig. 6.

## Discussion

We have shown that particles that are larger than a gel's characteristic mesh size are nevertheless able to facilitate their own diffusion through the gel by binding to crosslinking sites. Under the right conditions, this binding leads to local structural reorganization within the gel that allows binding particles to diffuse while non-binding particles remain caged. Importantly, the

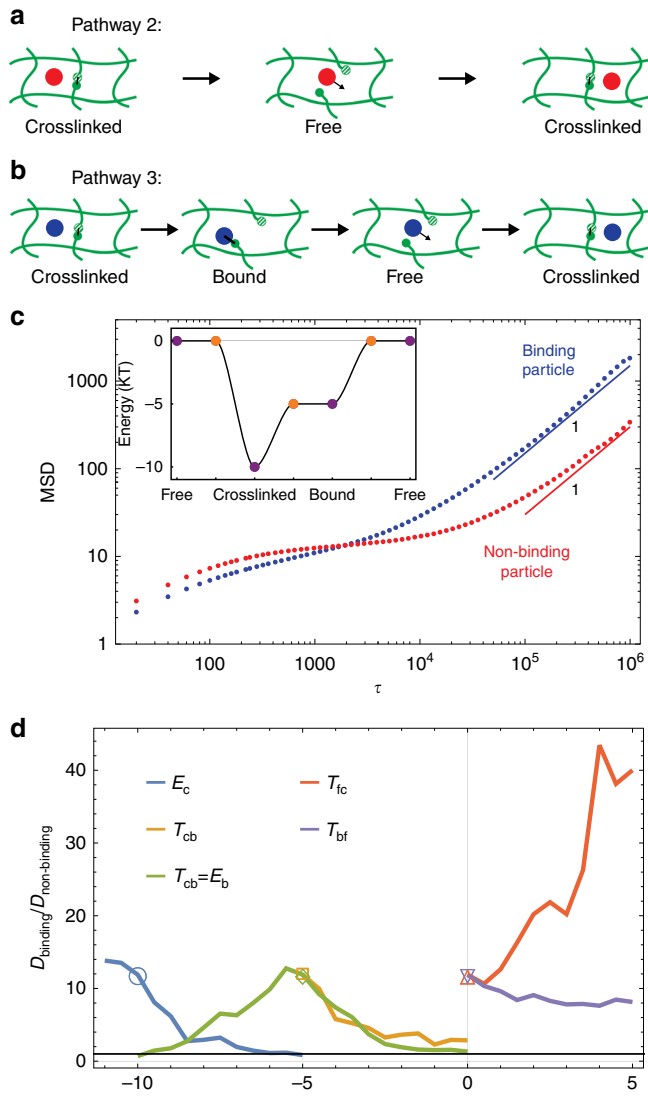

**Fig. 5** Enhanced diffusion in a reversible gel with moderate binding affinities. **a** A sketch of how a non-binding particle can diffuse when crosslinks are able to spontaneously break. **b** A sketch of a third pathway for diffusion that is only available to binding particles. **c** Mean squared displacement of a binding particle (blue) and a non-binding particle (red). At short time scales, the binding particle moves slower than the non-binding particle because it is bound. However, this binding helps break crosslinks on long-time scales and leads to a higher diffusion constant. See Supplementary Movie 3. **d** $D_{\text{binding}}/D_{\text{non-binding}}$ for five slices of the five-dimensional parameter space. Each curve gives $D_{\text{binding}}/D_{\text{non-binding}}$ while changing one or two of the energies while keeping all the others fixed. The symbols indicate the point on the curve corresponding to those initial energies. Data are averaged over 50 simulations in **c** and over 30 simulations for each point in **d**

mechanism outlined in this paper occurs in thermodynamic equilibrium: detailed balance is strictly satisfied and no energy is consumed. The proposed mechanism relies on two key features: (1) competitive binding, where the diffusing particle binds directly to the crosslink binding sites in such a way so that they cannot be both crosslinked and bound to the particle simultaneously; and (2) direct bond exchange, where the crosslink binding sites can transition directly from the crosslinked state to being bound to the particle without passing though the free state.

If a system does not exhibit some form of these key features, so that crosslink dynamics are influenced directly by particle binding, our mechanism will not be relevant.

Our results provide a microscopic mechanism that could potentially explain how the nuclear pore complex attains selectivity, but further work is required to verify this. One perplexing aspect of the NPC is that translocation is remarkably fast; Ribbeck and Gorlich[19] report a diffusion constant of $D \approx 0.23D_0$, where $D_0$ is the diffusion rate through an empty, plugless pore. In the high binding-affinity limit of our model, we are able to obtain a diffusion constant of $D \approx 0.059D_0$, where $D_0$ is the diffusion constant of a completely free particle with no gel, and the diffusion constant is proportional to the diffusion rate through a gel-filled pore. Our three state scheme (Fig. 2b) is surely less sophisticated than the crosslink dynamics of the NPC; nonetheless, it is reassuring that even such a simple scheme could achieve a diffusivity within a factor of 4 of the NPC. An interesting direction for future research is to explore the landscape of possible crosslink dynamics, to increase $D/D_0$.

The structural integrity of the polymer cage that surrounds the diffusing particles comes from a combination of crosslinks and entanglement. Since the mechanism presented here couples the crosslink dynamics to particle binding, we have focused on gels that are not entangled so as to most cleanly demonstrate the principle. As the degree of entanglement increases, the effectiveness of this mechanism should decrease, but as long as crosslinks have an integral role in holding together particle cages, enhanced diffusion should be possible. In addition, we have focused on equilibrium mechanisms for enhanced mobility, removing this constraint opens up a class of mechanisms (or variants of our mechanism) that could allow particles to diffuse faster than they would without any gel at all ($D > D_0$).

One important consideration that we did not fully explore is the dependence on the concentration of diffusing particles. When a binding particle passes the vicinity of a crosslink site in the moderate binding affinity limit, the statistics (e.g., probability of being in the crosslinked state) change. This effect decays over the self-healing time of the gel. Thus, if a non-binding particle encounters this region over this time scale, it can experience a slight enhancement in mobility. The case of high binding affinity presents an additional complication as two binding particles are able to rearrange the gel topology (which is usually static in this limit), which can lead to one of the binding particles getting stuck. This can be seen on the right-hand side in Supplementary Movie 2.

Finally, our results provide design rules for building artificial gels with selective permeability. While competitive binding and direct bond exchange are nontrivial requirements, there are a number of systems for which they are possible. For example, DNA-based gels are a particularly appealing model system for a few reasons. Not only is it straightforward to create gels out of star-shaped DNA strands with functionalized ends[37–39], but competitive binding and direct bond exchange are natural components of DNA bonds. In addition, DNA strands can be designed to have arbitrary binding energies, and transition state energies can be tuned with transition-mediating linker strands. Another system that naturally exhibits direct bond exchange is the class of plastics called vitrimers[40–42], which are covalently bound networks that exchange bonds via thermally activated reactions.

## Methods

**Numerical details**. We consider a gel composed of $N$ small star-shaped polymers, in a two-dimensional square box of linear length $L$ with periodic boundary conditions. Note that diffusion through a polymer gel is qualitatively different in two and three dimensions because in two dimensions there is no way to go around a linear filament. We are interested in understanding a mechanism for diffusion in three dimensions, but we are also interested in the limit where particles cannot move around filaments due to their size. In two dimensions, all particles are explicitly in this limit regardless of size. Thus, it is cleaner to study the qualitative features of this mechanism in two dimensions rather than three, and we do not need to worry about the size of the diffusing particles.

The polymers have binding sites on the end of each strand; crosslinking between the individual elements forms a highly connected gel, with low entanglement. This property is important as it allows the gel to easily remodel. The star-shaped polymers are composed of 9 monomers connected in a cross formation (see the shaded gray particles in Fig. 3a), so that there is a central monomer attached to 4 chains of 2 monomers each. The light green lines in Fig. 3a show permanent bonds holding the polymers together. The polymers make a gel by forming crosslinks (solid black lines) between the $G$ and $G'$ binding sites located at the strand ends (solid green disks). Note that in the simulations we make the binding sites self-complementary (i.e., $G = G'$), but this choice has no effect on the mechanism being studied. A particle in the gel (blue disk) can either bind or not bind to the gel. A binding particle can only bind to the strand ends (dashed black line).

We model the gel monomers as disks of radius $R_m$, and the particle as a disk of radius $R_p$. The interaction energy between two disks at a center-to-center distance $r_{ij}$ with radii $R_i$ and $R_j$ is given by

$$V_H\left(r_{ij}, b_{ij}\right) = \frac{k_H}{2} s_{ij}^2 \times \begin{cases} 1 & \text{if } s_{ij} \leq 0 \\ b_{ij} & \text{if } s_{ij} > 0, \end{cases} \quad (5)$$

where $s_{ij} = r_{ij} - (R_i + R_j)$, and $k_H$ sets the strength of the interaction. The bond parameter $b_{ij}$ is a binary variable: $b_{ij} = 1$ if there is a bond between the disks, resulting in a two-sided harmonic potential, and $b_{ij} = 0$, if there is no bond, resulting in a one-sided harmonic potential that is zero if the disks do not overlap. In addition, for consecutive monomers $h$, $i$, and $j$ in a polymer strand, we include bond-bending interactions, which take the form

$$V_{BB}\left(\theta_{hij}\right) = \frac{k_{BB}}{2}\left(\cos\theta_{hij} - 1\right), \quad (6)$$

where $\theta_{hij}$ is the angle formed by the bond between monomers $h$ and $i$ and the bond between monomers $i$ and $j$, and $k_{BB}$ sets the strength of the bond-bending interactions.

We simulate the overdamped Langevin equation,

$$\frac{d\mathbf{r}_i}{dt} = \frac{1}{\gamma_i}\mathbf{F}_i + \sqrt{\frac{2k_B T}{\gamma_i}}\mathbf{f}_i(t), \quad (7)$$

for disk $i$, where $\gamma_i$ is the friction coefficient, $\mathbf{F}_i$ is the total force on the disk (calculated by taking the gradient of the total energy), $k_B$ is the Boltzmann constant, $T$ is the temperature, and the elements of $\mathbf{f}_i(t)$ are Gaussian random variables that satisfy $\langle f_i^\alpha(t) \rangle = 0$ and $\langle f_i^\alpha(t) f_j^\beta(t') \rangle = \delta_{ij}\delta_{\alpha\beta}\delta(t - t')$, where $\alpha$, $\beta$ index spatial dimensions. The friction coefficient is given by $\gamma_i = 2d\pi\eta R_i$, where $d$ is the dimensionality of the system and $\eta$ is the dynamic viscosity. Equation (7) is integrated numerically using a fixed time step $\Delta t$. The numerical parameters are given in Table 1.

The vast majority of the bond variables $b_{ij}$ are fixed to either 0 or 1 and do not change over time. However, the $b_{ij}$'s between strand ends and between the strand ends and the binding particle can change in time, as discussed above. Importantly, a bond is only allowed to change if the two disks are overlapping (i.e., $s_{ij} < 0$). If a bond were to change while the disks were not overlapping, the energy in Eq. (5) would change discontinuously and detailed balance would not be satisfied. If the disks are overlapping, the transitions depicted in Fig. 2b occur with rate $k_{\mu\nu}$. In

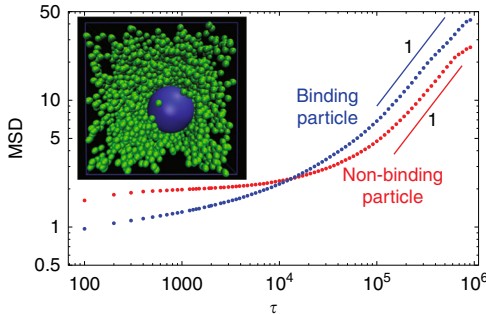

**Fig. 6** Enhanced diffusion in a three-dimensional reversible gel with moderate binding affinities. The mean squared displacement of a binding particle (blue) and a non-binding particle (red) shows the same general features as in two dimensions (Fig. 5c). Inset: snapshot of the diffusing particle and the gel

**Table 1 List of parameters in Brownian dynamics simulations**

| Parameter | Value |
|---|---|
| $R_m$ | 0.5 |
| $R_P$ | 1 |
| $k_H$ | 400 |
| $k_{BB}$ | 200 |
| $\eta$ | 1 |
| $k_B T$ | 1 |
| $\Delta t$ | 0.005 |

between each time step of the Brownian dynamics simulation, transitions are calculated via a Gillespie algorithm that terminates at time $\Delta t$.

**Code availability**. The computer code that supports the findings of this study are available from the corresponding author upon reasonable request.

## Data availability

The data that support the findings of this study are available from the corresponding author upon reasonable request.

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

## Acknowledgements

We thank Yohai Bar Sinai, Alpha Lee and Caroline Wagner for helpful discussions. This research was funded by the National Science Foundation through the Harvard Materials Research Science and Engineering Center Grant DMR-1420570, DMREF Grant DMR-123869 and ONR Grant N00014-17-1-3029. K.R. acknowledges funding from NIBIB/NIH Grant R01 EB017755-04, the National Science Foundation Career award PHY-1454673, and the MRSEC Program of the National Science Foundation under award DMR-1419807. M.P.B. is an investigator of the Simons Foundation.

## Author contributions

C.P.G., M.P.B. and K.R. designed research and wrote the paper. C.P.G. performed numerical simulations and analyzed data.

## Additional information

**Competing interests:** The authors declare no competing interests.

