## [Peer Review File · Nature Communications]

Reviewers' comments:

Reviewer #1 (Remarks to the Author):

This manuscript presents theoretical modeling of a physical mechanism of particle/gel crosslink binding to lead to enhanced particle transport within gels mesh size is smaller than the particle. Typically, when a particle radius of gyration is larger than the gel mesh size, diffusion within the gel is possible due to gel structure rearrangements but very slow. Here, the authors propose a new mechanism where a particle is able to competitively bind with a crosslink site resulting in a structural rearrangement that allows the binding particle to diffuse through the network, even when that particle is of the order of the mesh size, where a non-binding particle would remain entrapped.

Methods, interpretations and conclusions appear to be solid. The concept is logical and appears to be novel. The potential relevance of such a mechanism to enable controlled interaction filtering in gels and networks is clear for understanding biological processes as well as development of new technologies for fields such as drug delivery or tissue engineering

Questions:

1. It was not clear to me what the physical source of the maximum peaks were in Figure 5D for negative energies (e.g. green and blue lines). Both maximize at $D_{\text{binding}}/D_{\text{nonbinding}} \sim 12$ but one at $E = -5$ and one at $E = -12$. Could this be explained a bit more thoroughly in the text to help readers who are non-experts in this field? The phenomena occurring at positive energies in this plot are explained.
2. Can the authors comment on the concentration dependence of the model? Is this only valid for very dilute particles as depicted in the BD simulations (e.g. 1 particle for ~ 100 transient crosslinks)? Clearly with enough bound particles the integrity of the gel would be compromised similarly to the polymer degradation model discussed in Figure 2A.
3. Is there anything known about the exact chemical make-up of the "specific hydrophobic binding interactions" in NCPs discussed on page 2? Can one say anything about the strengths of these interactions and if they are at all close to matching the criteria required for accelerated transport in the proposed model (competitive binding as well as direct bond exchange)?
4. Also, it was not clear to me if there is any known transient crosslinking occurring in the NCP that would be comparable to the proposed model. My understanding was the nuclear pore complex consists of multiple proteins (FG Nups, etc) that particle exchange through the NCP is mediated by various transport factors. Could the authors clarify this? I'm wondering how appropriate the NCP comparisons are for this model.

Reviewer #2 (Remarks to the Author):

This is an interesting manuscript which analyzes an important issue -- how polymer gel can serve as a filter passing through some particles and not passing others. The problem is obviously relevant in both biological and chemical engineering settings. The problem is particularly difficult for relatively large particles, which are too big to fit in the gel mesh. The article is nicely and clearly written. It is based on a simple transparent physical idea: if the particle has the ability to realize bond-exchange reaction, then it can temporarily unlock some of the cross-links, pass through the open gate, and then close the door behind by the reverse bond-exchange. Authors present both some analytical arguments and (relatively simple) Brownian dynamics simulation to illustrate the enhanced diffusion of bond-exchanging particles in the bulk of the gel. I have three concerns regarding this work:

(1) I do not understand why concentration of diffusing particles is nowhere to be seen in the equations. I think in reality energy E_b must include chemical potential of these blue particles. That would imply that b state becomes more populated (the sum of two probabilities in Eq. 1 and 2 decreases) with increasing concentration.

(2) Authors concentrate exclusively on how their particles diffuse through the bulk of the gel. In reality, for any filter, be it an industrial filter or Nuclear Pore complex, particles have to enter the gel from one surface and leave through the other, and what counts is overall flux. It seems to me that by entering the gel and starting the bond exchange game, particle first of all dissipates some energy of the order of $150 \text{ k}_B T$ or maybe half that, based on the numbers used in simulations (e.g., caption of figure 3). This implies that exiting the gel particle will face a similar height barrier, essentially an insurmountable one. This seemingly will lead to huge accumulation of particles inside the gel, reducing energy E_b (see my point 1), and possibly alleviating the exit problem - although I do not see it.

To summarize my first two points, I think that the authors should really consider the situation in its entirety, including entrance and exit from the gel, and to look at either steady flux or mean first passage time.

(3) I do not see how opening the gate by bond exchanging particle will not allow other particles of a similar size to sneak in. Since the life time of crosslinked state decreases, it seems that permeation of competing 'wrong' particles should increase.

To summarize, this is an interesting thought provoking study. I would be glad to enthusiastically support its acceptance and publication provided that the authors explain away my concerns.

Reviewer #3 (Remarks to the Author):

In this contribution, the authors propose a mechanism of competitive bindings that promotes enhanced diffusion of a particle through a gel network.

This interesting claim is exemplified for a simple two-dimensional model where parameter combinations are identified for this mechanism to work.

While the topic is certainly interesting for readers of Nature, the manuscript in its present form does not seem to be fully convincing as detailed below.

1. My first main comment concerns the case of high binding affinity - the main result of this study. While it is plausible to consider the case where bonds are very unlikely to be spontaneously broken by thermal energy but can switch to a neighboring particle, it is much less clear why such a switch should be forbidden to occur to other monomer particles when they are spatially close. After all, these particles also have a high binding affinity to each other so that they form and re-organize the network (see e.g. Yan et al, Scientific Reports 2016 or Amin et al, Macromolecules 2016 on dynamics of telechelic polymers).

2. The main drawback of the work seems to be its limitation to a strictly two-dimensional case, both in terms of the simulation study and the heuristic argument. Obviously, crossing a single strand is sufficient for a particle to cross to a neighboring pore in two spatial dimensions. However, the situation is quite different in three dimensions since trapping a particle in three dimensions by one-dimensional strands requires a more complicated topology. It is unclear to me whether in a random network that is highly crosslinked (to be able to trap the particle) opening single strands would be sufficient for enhanced diffusion. The situation seems unclear even for the highly idealized case of a perfectly simple cubic network, where one would need either rather stiff strands

or mesh sizes that are significantly smaller than the particle diameter for efficient trapping.

3. Do Eqs 1 and 2 refer to the case where direct transitions from crosslinked to bound states are forbidden? Otherwise, k_{cb} and k_{bc} should appear in these expressions. Some clarifications are necessary as this is potentially confusing for readers.

4. On page 3, the authors state a priori constraints on the model parameters $T_{cb} > E_b$. But actually the authors choose $T_{cb} = E_b$, see caption of Fig. 3.

Reviewer #1:

This manuscript presents theoretical modeling of a physical mechanism of particle/gel crosslink binding to lead to enhanced particle transport within gels mesh size is smaller than the particle. Typically, when a particle radius of gyration is larger than the gel mesh size, diffusion within the gel is possible due to gel structure rearrangements but very slow. Here, the authors propose a new mechanism where a particle is able to competitively bind with a crosslink site resulting in a structural rearrangement that allows the binding particle to diffuse through the network, even when that particle is of the order of the mesh size, where a non-binding particle would remain entrapped.

Methods, interpretations and conclusions appear to be solid. The concept is logical and appears to be novel. The potential relevance of such a mechanism to enable controlled interaction filtering in gels and networks is clear for understanding biological processes as well as development of new technologies for fields such as drug delivery or tissue engineering

Questions:

1. It was not clear to me what the physical source of the maximum peaks were in Figure 5D for negative energies (e.g. green and blue lines). Both maximize at D binding/D nonbinding ~ 12 but one at $E = -5$ and one at $E = -12$. Could this be explained a bit more thoroughly in the text to help readers who are non-experts in this field? The phenomena occurring at positive energies in this plot are explained.

The referee is correct in pointing out that we did not thoroughly discuss all the features of Fig. 5D in the text. The green line has a maximum for the following reason: at the left end of the curve ($E_b=T_{cb}=E_c$, and $T_{fc}=T_{fb}=0$), the rate of getting to the free state is the same when coming from the crosslinked state as when coming from the bound state. In other words, being in the bound state neither helps nor hurts you, so D binding/D nonbinding approaches 1. Similarly, when $E_b=T_{cb}=0$, the

rate of going from the crosslinked to bound state is the same as crosslinked to free. In the intermediate region, the bound state acts as a catalyst to get from the crosslinked to the free state, so $D_{\text{binding}} > D_{\text{nonbinding}}$.

For the blue line, which explores the effect of changing E_c : we were initially unable to explain the decrease in $D_{\text{binding}}/D_{\text{nonbinding}}$ at very low E_c , but upon further investigation we discovered a numerical issue having to do with the proper equilibration of some of the simulations at the lowest energies ($E_c < -11$). Therefore, we cannot say if there in fact is a maximum or not. We have checked thoroughly that all other data was properly equilibrated and are grateful to the referee for prompting us to perform these additional checks.

2. Can the authors comment on the concentration dependence of the model? Is this only valid for very dilute particles as depicted in the BD simulations (e.g. 1 particle for ~ 100 transient crosslinks)? Clearly with enough bound particles the integrity of the gel would be compromised similarly to the polymer degradation model discussed in Figure 2A.

The referee is correct that we did not systematically explore the effect of concentration in the present manuscript, and that most of our simulations are in the low-concentration limit.

In the limit of high-binding affinity, Fig. 4B and Movie S2 show a simulation at higher concentration. There are two ways in which finite concentrations can affect diffusion here: 1) When two binding particles are in the same vicinity, they can alter the crosslink topology (for example, if both are in the bound state and one G-binding site uses the other's G'-binding site to reform a crosslink), which can lead to a binding particle getting stuck. This is actually observed in Movie S2 on the right-hand side. 2) There is also clearly the possibility that a binding particle could open one of the crosslinks forming the cage of a non-binding particle, allowing the non-binding particle to diffuse a short distance. While this is unavoidable given the nature of the model, we note that it is not observed in Movie S2, although we do observe it at higher concentrations.

In the limit of moderate binding affinity, the way to think about the concentration dependence is to consider the effect that a single binding particle has on the structure of the gel. Whatever effect a passing particle has on the local crosslink statistics will decay over the "self-healing" time scale (the time scale over which the gel naturally reorganizes its structure). The relevant quantity is the likelihood that a non-binding particle will encounter crosslinks whose statistics are still feeling the effect of the binding particle. When this is the case, the non-binding particle will experience a slight enhancement in mobility. The details of this depend on a number of things, including the microscopic implementation of the gel, and a systematic study is beyond the scope of the present manuscript.

We have added a paragraph in the discussion of the main text that summarizes this.

3. Is there anything known about the exact chemical make-up of the “specific hydrophobic binding interactions” in NCPs discussed on page 2? Can one say anything about the strengths of these interactions and if they are at all close to matching the criteria required for accelerated transport in the proposed model (competitive binding as well as direct bond exchange)?

The specific binding within the NPC is due to hydrophobic interactions related to the FG domains [e.g. it is known that the interaction between tryptophan on the transporter protein NTF2 and the FG repeats units is necessary for efficient transport, see Bayliss et al. EMBO J 21 2843 (2002) and Bayliss et al. JMB 293 579 (1999)]. While binding energies are difficult to measure directly, the dissociation constant has been measured to be in the micro-molar range [$K_d > 4\mu\text{M}$, Ribbeck and Gorlich, EMBO J 20 1320 (2001)], which implies low binding affinities.

Being more precise is difficult without specifying specific nuclear pore and transporter molecules. However, hydrophobic interactions are typically on the order of a few kT, meaning that the proper comparison is within the moderate binding affinity limit in our model. (In addition, Calwell et al. [PLoS Comput Biol. 6 (2010)] calculate electrostatic interactions that are also on the order of a few kT.)

4. Also, it was not clear to me if there is any known transient crosslinking occurring in the NCP that would be comparable to the proposed model. My understanding was the nuclear pore complex consists of multiple proteins (FG Nups, etc) that particle exchange through the NCP is mediated by various transport factors. Could the authors clarify this? I'm wondering how appropriate the NCP comparisons are for this model.

We do feel the NPC is a proper comparison. The FG nups within the pore have experimentally been shown to self-assemble into reversibly crosslinked hydrogels through hydrophobic interactions [Frey and Gorlich, Cell 130 512 (2007) and Frey et al. Science 314 815 (2006)]. Furthermore, these FG-nup gels form highly selective barriers, allowing passage only of transport receptors (and their complexes) which have the ability to bind to FG-domains, while preventing transport of inert particles which cannot interact with FG domains. This selective solvation of nuclear transport receptors through the nuclear pore complex is paradoxical because their binding to the FG-repeats should delay passage; instead, this binding interaction facilitates diffusion. [Frey and Gorlich, Cell 130 512 (2007) and Frey et al. Science 314 815 (2006)]. Our work suggests a mechanistic explanation for how such reversibly crosslinked network can act as a selective filter.

Reviewer #2 (Remarks to the Author):

This is an interesting manuscript which analyzes an important issue -- how polymer gel can serve as a filter passing through some particles and not passing others. The problem

is obviously relevant in both biological and chemical engineering settings. The problem is particularly difficult for relatively large particles, which are too big to fit in the gel mesh. The article is nicely and clearly written. It is based on a simple transparent physical idea: if the particle has the ability to realize bond-exchange reaction, then it can temporarily unlock some of the cross-links, pass through the open gate, and then close the door behind by the reverse bond-exchange. Authors present both some analytical arguments and (relatively simple) Brownian dynamics simulation to illustrate the enhanced diffusion of bond-exchanging particles in the bulk of the gel. I have three concerns regarding this work:

(1) I do not understand why concentration of diffusing particles is no-where to be seen in the equations. I think in reality energy E_b must include chemical potential of these blue particles. That would imply that b state becomes more populated (the sum of two probabilities in Eq. 1 and 2 decreases) with increasing concentration.

The referee is correct that the crosslink dynamics do indeed depend on the concentration of the diffusing particles (as well as the concentration of the crosslink binding sites), but it is actually local, not global, concentrations that matter. In our original manuscript we had absorbed this effect into the rate constants in Equations 1-3, but we now realize that our explanation was not sufficiently clear. We have rewritten this section to explicitly include the effects of concentrations and feel it is now much more clear.

(2) Authors concentrate exclusively on how their particles diffuse through the bulk of the gel. In reality, for any filter, be it an industrial filter or Nuclear Pore complex, particles have to enter the gel from one surface and leave through the other, and what counts is overall flux. It seems to me that by entering the gel and starting the bond exchange game, particle first of all dissipates some energy of the order of $150 k_B T$ or maybe half that, based on the numbers used in simulations (e.g., caption of figure 3). This imply that exiting the gel particle will face a similar height barrier, essentially an insurmountable one. This seemingly will lead to huge accumulation of particles inside the gel, reducing energy E_b (see my point 1), and possibly alleviating the exit problem -- although I do not see it.

The referee is correct that what counts is the overall flux, and that the details of how the particle enters and exits the gel play a role. However, the particles do not dissipate $150k_B T$ of energy when they enter the gel and, more importantly, do not have to overcome such a barrier when they exit. The reason for this is that when the particle forms a bond with one of the crosslink binding sites, the corresponding crosslink breaks simultaneously. The full $150k_B T$ energy barrier associated with spontaneously breaking a bond never has to be overcome because the direct path between the bound and crosslinked states has a much lower energy barrier of only a few $k_B T$ (Fig. 3B-inset). Figure 4B and Movie S2 show that particles are indeed able to exit a gel even in the limit of high binding affinity.

To summarize my first two points, I think that the authors should really consider the situation in its entirety, including entrance and exit from the gel, and to look at either steady flux or mean first passage time.

With regards to considering the situation in its entirety, the referee is correct that crossing a gel barrier is a three step process: the particle enters the gel, diffuses through it, and then exits on one side or the other. One can imagine a filter that differentiates between binding and non-binding particles in any one of these three steps. However, the primary scientific contribution of our paper is to demonstrate a mechanism for enhanced mobility, meaning that the differentiation occurs primarily in the second step, and so the most direct way to test the effectiveness of this mechanism is to focus on the diffusivity.

Nevertheless, it is still important to consider the situation in its entirety to demonstrate that the full system can indeed act as a filter. This is demonstrated in Figure 4B and Movie S2, where binding particles are clearly able to enter the gel, diffuse through it, and exit on the other side while non-binding particles are completely retained on the original side of the barrier.

(3) I do not see how opening the gate by bond exchanging particle will not allow other particles of a similar size to sneak in. Since the life time of crosslinked state decreases, it seems that permeation of competing 'wrong' particles should increase.

The referee is correct that the diffusion constant of non-binding particles technically increases slightly due to the decrease in the lifetime of the crosslinks that is caused by the binding particles. However, this is a local and transient effect (only the crosslinks near the binding particle experience this change in lifetime). Therefore, non-binding particles only experience this effect when very close to a binding particle, and the net effect on non-binding particles vanishes in the dilute limit. Nevertheless, this effect does not vanish at finite concentrations of diffusing particles, see discussion in response to referee 1. We have added a paragraph in the discussion that addresses this.

To summarize, this is an interesting thought provoking study. I would be glad to enthusiastically support its acceptance and publication provided that the authors explain away my concerns.

Reviewer #3 (Remarks to the Author):

In this contribution, the authors propose a mechanism of competitive bindings that promotes enhanced diffusion of a particle through a gel network. This interesting claim is

exemplified for a simple two-dimensional model where parameter combinations are identified for this mechanism to work. While the topic is certainly interesting for readers of Nature, the manuscript in its present form does not seem to be fully convincing as detailed below.

1. My first main comment concerns the case of high binding affinity - the main result of this study. While it is plausible to consider the case where bonds are very unlikely to be spontaneously broken by thermal energy but can switch to a neighboring particle, it is much less clear why such a switch should be forbidden to occur to other monomer particles when they are spatially close. After all, these particles also have a high binding affinity to each other so that they form and re-organize the network (see e.g. Yan et al, Scientific Reports 2016 or Amin et al, Macromolecules 2016 on dynamics of telechelic polymers).

The referee is correct that one could imagine including transitions where a G binding site that is initially bound to a particular G' site (forming a crosslink) could transition directly to being bound to a different G' site, and that such a transition would allow for diffusion of a non-binding particle even in the high affinity limit. However, there is no reason that the transition state energy governing the rate of such a transition should be the same as that governing the crosslinked-to-bound transition. We consider the simplest case where this energy barrier is arbitrarily high, but point out in the discussion that an interesting direction for further work is to explore models with more complicated kinetic diagrams, which includes this case.

Perhaps a more pressing question is whether or not this limit that we consider is useful for real systems, and we believe strongly that it is. For example, DNA based gels allow one to control precisely what transitions are allowed by mediating the bonds with linker strands. For the Nuclear Pore Complex, the moderate affinity limit is likely more relevant than the high affinity limit, and in this case the crosslink-to-crosslink transitions that the referees propose will have much smaller affect on the relative dynamics.

2. The main drawback of the work seems to be its limitation to a strictly two-dimensional case, both in terms of the simulation study and the heuristic argument. Obviously, crossing a single strand is sufficient for a particle to cross to a neighboring pore in two spatial dimensions. However, the situation is quite different in three dimensions since trapping a particle in three dimensions by one-dimensional strands requires a more complicated topology. It is unclear to me whether in a random network that is highly crosslinked (to be able to trap the particle) opening single strands would be sufficient for enhanced diffusion. The situation seems unclear even for the highly idealized case of a perfectly simple cubic network, where one would need either rather stiff strands or mesh sizes that are significantly smaller than the particle diameter for efficient trapping.

We appreciate the referees concern regarding the qualitative difference between diffusing through a gel in two and three dimensions. While there are valid reasons (discussed in the first paragraph of the “Methods” section) that two-dimensional simulations present a cleaner demonstration of the mechanism in question, we agree that it is still important to demonstrate that this works in three dimensions as well. To address this concern, we have performed 3d simulations in the moderate binding affinity limit and verified that enhanced diffusion is still obtained. This is now discussed in the text and in Figure 6.

3. Do Eqs 1 and 2 refer to the case where direct transitions from crosslinked to bound states are forbidden? Otherwise, k_{cb} and k_{bc} should appear in these expressions. Some clarifications are necessary as this is potentially confusing for readers.

These equations do indeed refer to the case where direct bond exchange is forbidden -- we have rewritten this section and feel that this is now more clear. However, we note that equilibrium concentrations do not depend on the topology of kinetic diagrams and so these equations would actually not be affected if direct bond exchange were allowed (although one *could* write them in terms of k_{cb} and k_{bc} if one so desired).

4. On page 3, the authors state a priori constraints on the model parameters $T_{cb} > E_b$. But actually the authors choose $T_{cb} = E_b$, see caption of Fig. 3.

We thank the referee for catching this, the constraints should be that $T_{cb} \geq E_b$, etc. This is now fixed in the text.

REVIEWERS' COMMENTS:

Reviewer #1 (Remarks to the Author):

The revised manuscript and comments to the editor have sufficiently addressed my questions/comments. I support the publication of the revised manuscript in its current form.

Reviewer #2 (Remarks to the Author):

In my opinion, the manuscript is significantly improved. I am positively inclined, because the work deals with a question of selective filtering which is of great significance, and offers an original solution. It seems to me that authors in the revised version satisfactorily addressed most of the critical remarks.

To me, there is only one concern. The whole idea is based on the assumption that two energies very accurately match and balance one another; this is how authors explain why the particle entering the gel does not dissipate energy of the order of 100 k_BT . This is because, if I understood correctly, the particle binds with energy $E_1 \sim 100 \text{ k}_\text{BT}$, but does so on the expense of opening a gate which itself requires energy $E_2 \sim 100 \text{ k}_\text{BT}$. Thus, the idea is that both energies E_1 and E_2 are huge, but their difference is modest, on the order of 1 k_BT . In this form it is nice, but how can it be realized in nature? For example, if particles in question are proteins (or RNA), then energy E_1 is the sum of several contributions due to contacts by individual aminoacids (or nucleotides). How can one imagine a mechanism that could control this sum to within 1% ? Another view of the same problem is this. Exchange reaction of the necessary type can be thought of as a nano-scale machine which has to take energy E_1 and transform it to E_2 mechanically, i.e., without any loss (or about 1% loss) to heat. How can one imagine a nano-scale machine with such efficiency?

I agree with the authors that their idea is interesting and exciting and I would be satisfied if the authors admit in the text that their idea does face some difficulties (like the ones outlined above) which authors hope to resolve in the future. But I think the reader deserves to be warned that such difficulties exist -- and the work will be actually better if difficulties are discussed.

Reviewer #3 (Remarks to the Author):

The authors have addressed all my comments in a satisfactory manner. I am therefore happy to recommend publication.

September 19, 2018

Dear Editors and Reviewers,

We are delighted that all the reviewers responded positively to our revised manuscript and would like to sincerely thank them for their hard work. Reviewer #2 had one additional concern, which we have addressed. The Reviewer comments are below in blue and our response to Reviewer #2 is in bold.

REVIEWERS' COMMENTS:

Reviewer #1 (Remarks to the Author):

The revised manuscript and comments to the editor have sufficiently addressed my questions/comments. I support the publication of the revised manuscript in its current form.

Reviewer #2 (Remarks to the Author):

In my opinion, the manuscript is significantly improved. I am positively inclined, because the work deals with a question of selective filtering which is of great significance, and offers an original solution. It seems to me that authors in the revised version satisfactorily addressed most of the critical remarks.

To me, there is only one concern. The whole idea is based on the assumption that two energies very accurately match and balance one another; this is how authors explain why the particle entering the gel does not dissipate energy of the order of 100 k_BT . This is because, if I understood correctly, the particle binds with energy $E_1 \sim 100 \text{ k}_\text{BT}$, but does so on the expense of opening a gate which itself requires energy $E_2 \sim 100 \text{ k}_\text{BT}$. Thus, the idea is that both energies E_1 and E_2 are huge, but their difference is modest, on the order of 1 k_BT . In this form it is nice, but how can it be realized in nature? For example, if particles in question are proteins (or RNA), then energy E_1 is the sum of several contributions due to contacts by individual aminoacids (or nucleotides). How can one imagine a mechanism that could control this sum to within 1% ? Another view of the same problem is this. Exchange reaction of the necessary type can be thought of as a nano-scale machine which has to take energy E_1 and transform it to E_2 mechanically, i.e., without any loss (or about 1% loss) to heat. How can one imagine a nano-scale machine with such efficiency?

I agree with the authors that their idea is interesting and exciting and I would be satisfied if the authors admit in the text that their idea does face some difficulties

(like the ones outlined above) which authors hope to resolve in the future. But I think the reader deserves to be warned that such difficulties exist -- and the work will be actually better if difficulties are discussed.

We have added the following sentences to the last paragraph in the “Perfect filtering with high binding affinity” section, which already discussed one other difficulty with this limit. “Another difficulty when considering this high affinity limit is that E_c and E_b must be matched to within a few $k_{\mathrm{B}}T$ despite the magnitude of the energies being considerably larger. Whether in natural or artificial gels, it is not always clear how such precise control of the binding energies can be obtained.”

Reviewer #3 (Remarks to the Author):

The authors have addressed all my comments in a satisfactory manner. I am therefore happy to recommend publication.